# Estimation of Linkage Disequilibrium and Effective Population Size in Three Italian Autochthonous Beef Breeds

**DOI:** 10.3390/ani10061034

**Published:** 2020-06-14

**Authors:** Maria Chiara Fabbri, Christos Dadousis, Riccardo Bozzi

**Affiliations:** Dipartimento di Scienze e Tecnologie Agrarie, Alimentari, Ambientali e Forestali, Università di Firenze, 50144 Firenze, Italy; christos.dadousis@unifi.it (C.D.); riccardo.bozzi@unifi.it (R.B.)

**Keywords:** linkage disequilibrium, conservation, effective population size, local breeds

## Abstract

**Simple Summary:**

Linkage disequilibrium (LD) of genomic markers is related to various evolutionary forces, such as inbreeding, nonrandom mating, population bottleneck, drift, recombination, and mutations, and hence is an essential parameter to examine population history. In this analysis, we examined the LD pattern of three Italian local beef breeds (Calvana, Mucca Pisana, and Pontremolese) facing the risk of extinction, using the commercial Limousin beef breed as a control. Our results provide important information on the population history and the current status of the breeds and they can be further used for conservation and breeding purposes.

**Abstract:**

The objective was to investigate the pattern of linkage disequilibrium (LD) in three local beef breeds, namely, Calvana (n = 174), Mucca Pisana (n = 270), and Pontremolese (n = 44). As a control group, samples of the Italian Limousin breed (n = 100) were used. All cattle were genotyped with the GeneSeek GGP-LDv4 33k SNP chip containing 30,111 SNPs. The genotype quality control for each breed was conducted separately, and SNPs with call rate < 0.95 and minor allele frequency (MAF) > 1% were used for the analysis. LD extent was estimated in PLINK v1.9 using the squared correlation between pairs of loci (*r*^2^) across autosomes. Moreover, *r*^2^ values were used to calculate historical and contemporary effective population size (*N_e_*) in each breed. Average *r*^2^ was similar in Calvana and Mucca Pisana (~0.14) and higher in Pontremolese (0.17); Limousin presented the lowest LD extent (0.07). LD up to 0.11–0.15 was persistent in the local breeds up to 0.75 Mbp, while in Limousin, it showed a more rapid decay. Variation of different LD levels across autosomes was observed in all the breeds. The results demonstrated a rapid decrease in *N_e_* across generations for local breeds, and the contemporary population size observed in the local breeds, ranging from 41.7 in Calvana to 17 in Pontremolese, underlined the demographic alarming situation.

## 1. Introduction

In recent years, there has been a greater interest in recovering and preserving local breeds, especially for their adaptation’s capacity in marginal areas and for their importance as reservoir of genetic diversity.

In this context, Tuscany (central region of Italy) represents an important pool of genetic diversity with six different cattle breeds native from this area [1]. Three of these six breeds (Calvana (CAL), Mucca Pisana (MUP), and Pontremolese (PON)) have been recognized from the Italian breeders Association (AIA; Associazione Italiana Allevatori, Rome) to be at risk of extinction and at present are enrolled to the register of cattle breeds at limited diffusion (Registro Anagrafico delle razze bovine autoctone a limitata diffusione). From those, PON faces a critical risk of extinction, consisting a limited number of animals (n = 49). Historically, PON originated from the provinces of Massa Carrara, La Spezia, and Parma and, in the past, was used for the transport of Carrara marble, being a robust and rustic breed. CAL (n = 366) originates from the Calvana mountain, in Prato’s province. It is particularly suitable to live in marginal areas and has ever been considered as a beef breed. MUP (n = 413) is mainly reared nowadays in the province of Pisa. The breed is a crossbreed, mainly between Schwyz and Chianina breeds [2,3]. Hence, the historical, cultural, and ecological values of these three breeds are enormous and undisputed. Furthermore, the landscape of Tuscany placed barriers on widespread intensive breeding. As a result, the conservation and increase in sample sizes of local breeds could be of economic importance for this region.

Exploring genetic diversity is essential for developing conservation programs in autochthonous breeds, and one of the most commonly used parameters to assess genetic diversity is the effective population size (*N_e_*) [4]. *N_e_* is defined as the size of an ideal population that explains the same rate of random genetic changes as the current population [5,6], and has been traditionally estimated from the pedigree. However, with the advent of genomic technology, *N_e_* can also be inferred from genomic data. The pedigree-based method requires an adequate completeness of data over several generations [7], and for local breeds, this is not an easy task for practical reasons related to breeding management. In a previous study [8], *N_e_* from pedigrees was calculated for the same Tuscan breeds, but generally, the low quality of pedigrees and the alarming demographic situation of these breeds have raised the necessity to estimate the effective population size with genomic data; the linkage disequilibrium (LD) method was chosen for this purpose.

LD of genomic markers is related to various evolutionary forces, such as inbreeding, nonrandom mating, population bottleneck, drift, recombination, and mutation, and hence is an essential parameter to examine population history and genetic diversity. LD is defined as the nonrandom association between alleles at two (or more) loci [9]. In general, it is expected that the strength of LD decreases with an increased distance between markers located on the same chromosome. The term linkage disequilibrium (also known as gametic disequilibrium) goes back to Lewontin and Kojima, in 1960 [9,10]. The interest in studying LD patterns grew together with the research in genes associated with diseases [11,12]. Moreover, LD evaluation is an important prerequisite in genome wide association studies (GWAS), useful to detect the number of markers that will be sufficient for quantitative trait locus mapping [13,14]. Information of LD has been utilized in genomic breeding programs, such as marker-assisted selection and whole genome predictions [15]. The cost reduction and the efficient implementation of genotyping in animal breeding have made LD a common analysis in various species, such as pigs [16,17], poultry [18], sheep [19], goats [20], and cattle [21,22,23]. The most common LD measures are the squared genetic correlation coefficient (*r*^2^) described by Hill and Robertson [24] and D’ reported by Lewontin [10]. LD patterns in livestock populations have been analyzed to investigate (i) the structure and the history of populations [25], (ii) gene mapping [26], and (iii) the effective population size (*N_e_*) with molecular data [7,27,28].

The aim of this study was to investigate the LD and the LD-based *N_e_* patterns of three Italian beef cattle breeds (CAL, MUP, and PON) facing risk of extinction. The commercial Limousin beef breed (LIM) was included in the analysis to allow for comparisons between local unselected breeds and a cosmopolitan counterpart.

## 2. Materials and Methods

### 2.1. Animals and Sampling

A total of 588 beef cattle from four breeds (CAL = 174, MUP = 270, PON = 44, and LIM = 100) were genotyped. The percentage of the sampled cattle relative to the entire pool per breed for the local breeds was 47.5%, 65.4%, and 89.8% for CAL, MUP, and PON, respectively. Regarding the LIM, a sample of 100 cattle was extracted at random from a pool of 533 genotyped cattle, belonging to the last three generations and balanced by sex (52 males and 48 females). Genotypic data from LIM were provided by ANACLI (Associazione nazionale allevatori delle razze bovine Charolaise e Limousin, Roma) [29].

### 2.2. Genotyping and Quality Control

All cattle were genotyped with GeneSeek GGP-LDv4 33k (Illumina Inc., San Diego, CA, USA). Genotype quality control (QC) and data filtering were performed with PLINK v1.9 [30], and was conducted separately for each breed. Only SNPs located on the 29 autosome chromosomes were included (n = 28,289). SNPs with minor allele frequency (MAF) lower than 1%, and with call rate <0.95 were removed. Further, SNPs with more than 10% missingness values and deviated from Hardy Weinberg Equilibrium (*p* < 0.000001), as well as animals with more than 10% missingness, were also removed. After filtering, 164, 263, 41, and 100 cattle and 23,646, 23,436, 22,791, and 23,279 SNPs remained for CAL, MUP, PON, and LIM, respectively (Table 1).

### 2.3. Genomic Relationship Matrix

The genomic relationship matrix (GRM) was created per breed to (i) investigate the among breed identical by state relationships and (ii) compare the status of the genotypic samples among the populations under study. The GRM was calculated with the following formula by VanRaden [31]:(1)$$GRM=\frac{ZZ{}^{\prime }}{2\sum p_{i}\left(1-p_{i}\right)}$$
where ***Z*** is a centered matrix of marker genotypes of all individuals and *p_i_* is the frequency of the second allele at locus *i.*
***Z*** was calculated from genotypes of reference population subtracting 2*p_i_* from the matrix X that defines the genotypes for each individual as 0, 1 or 2. Heatmap graphs for the GRM of each breed were produced in R software [32].

For each GRM, the following parameters were taken into account: (i) the mean of the diagonal values; (ii) the mean of all the off-diagonal; (iii) the minimum and maximum of the diagonal values; and (iv) the minimum and maximum of the off-diagonal values. For all the above-mentioned parameters, the absolute value and squared root were also estimated.

### 2.4. Linkage Disequilibrium

Linkage disequilibrium was measured using *r*^2^, which is the squared correlation of the alleles at two loci [24]. The *r*^2^ is considered to be a better measure of LD than *D’* because it is more robust and less sensitive to changes in effective population size and gene frequency [33,34]. The *r*^2^ ranges between 0 and 1 and was calculated as follows:(2)$$r^{2}=\frac{{\left(freq\;\left(AB\right)\times freq\;\left(ab\right)-freq\;\left(Ab\right)\times freq\;\left(aB\right)\right)}^{2}}{\left(freq\;\left(A\right)\times freq\;\left(a\right)\times freq\;\left(B\right)\times freq\;\left(b\right)\right)}$$
where *freq* (*A*), *freq* (*a*), *freq* (*B*), and *freq* (*b*) are the allele frequencies and *freq* (*AB*), *freq* (*ab*), *freq* (*Ab*), and *freq* (*aB*) are the genotype frequencies. The LD extent was calculated for all SNPs pairs of each chromosome using PLINK v1.9 [30] under the command: *--r*2 *–ld-window 99999 --ld-window-r*2 *0*, in order to take an interval less than 99,999 SNPs and to save in the output all SNPs pairs.

The LD decay was analyzed in order to compare differences between and within breeds: (i) LDs were binned into four intervals of 0.25 Mbp (0–0.25, 0.25–0.5, 0.5–0.75, and 0.75–1 Mbp), the mean and the standard deviation (SD) of *r*^2^ values were computed for each interval; (ii) LD was also investigated for each autosome, considering windows of 1 Kbp. The LD-decay plots were visualized using the *ggplot2* R package [35].

### 2.5. Estimation of Historical and Contemporary Effective Population Size

The historical and recent *N_e_* for all breeds were estimated with the SNeP software [36], which is based on the relationships between LD, *N_e_*, and the recombination rate. The default options were used. *N_e_* was analyzed starting 13 generations ago because the default maximum distance in SNeP was 4000 Kbp. High LD in closely linked SNPs reflects ancient population history (50 Kbp ≈ 1500 generations ago), while high LD between distant SNPs describes more recent history (4000 Kbp ≈ 12.5 generations ago) [37]. The following formula was used to estimate *N_e_* from LD [28]:(3)$$N_{e\left(t\right)}=\frac{1}{\left(4f\left(c_{t}\right)\right)}\;\left(\;\frac{1}{E[r_{adj}^{2}\left|c_{t}\right|}-\alpha \right)$$
where *N_e_*_(*t*)_ is the effective population size estimated for *t* generation ago, which is calculated as $t=1/\left(2f(c_{t}\right))$ [38]; *c_t_* is the recombination rate at *t* generations ago, defined for *a* specific physical distance between markers; *r*^2^*_adj_* is the LD estimate adjusted for sample size (164, 263, 41, and 100 for CAL, MUP, PON, and LIM, respectively); and *α* is a constant and set to 1, as suggested by Ohta and Kimura [39]. The contemporary effective population size (*cNe*) was calculated with NEESTIMATOR v.2 [40] with the mating model set to random; *cNe* means that the results referred to the time period of the sample size included in the analysis.

## 3. Results

### 3.1. Quality Control and Genomic Relationship Matrix

Table 1 summarizes the number of SNPs and individuals from each breed, before and after the quality control.

The GRM heatmaps per breed are presented in Figure 1. Individuals were ordered by farm to consider the farm management. Diagonal blocks, indicating highly related individuals, were mainly found for CAL and MUP. Moreover, this grouping was mainly attributed to the farm level. The off-diagonal values among the different farms indicate that there is reduced gene-flow among the farms, with the exception of CAL. In CAL, higher levels of relationship between individuals both between and within farms were observed. The lowest relatedness average between animals was found for LIM.

GRM summary statistics for each breed are reported in Appendix A. The mean of the diagonal value was <1 in all breeds. The average of the diagonal values was 0.99 for CAL, MUP, and LIM, and 0.97 for PON. The highest diagonal values were 1.75, 1.72, 1.54, and 1.22 for MUP, PON, CAL, and LIM, respectively. The minimum diagonal values ranged from 0.67 (MUP) to 0.78 (LIM). The average of off-diagonal was negative for all breeds. The highest off-diagonal maximum was found in MUP (1.09), followed by PON (0.84) and CAL (0.79), while the lowest was found in LIM (0.50).

### 3.2. Linkage Disequilibrium

The total autosomal length in the analyzed SNP chip was 2512 Mbp, with the shortest *Bos taurus* autosome (BTA) being the BTA25 (~42.8 Mbp) and the longest being the BTA1 (~158.5 Mbp). The dimensions of each chromosome, the number of SNPs, the mean distance between SNPs, and the longest interval between pairwise SNPs for each chromosome are shown in Appendix A. In all breeds analyzed, the highest number of SNPs was found on BTA1 (ranging from 1328 to 1282, in CAL and PON, respectively), and the smallest was found in BTA27 (from 451 to 435 SNPs in CAL and PON, respectively). The average distances between adjacent SNPs were 0.105, 0.106, and 108 ± 0.08 Mbp in CAL and MUP, LIM, and PON, respectively. The largest distances between two adjacent SNPs were found on BTA12, that is, 4.44, 2.89, 2.87, and 2.09 Mbp in LIM, CAL, MUP, and PON, respectively.

The average of the highest *r*^2^ values was found in PON (0.17), and the lowest in LIM (0.07), with CAL and MUP (0.14) at intermediate values. The mean and SD of *r*^2^ per chromosome and breed are summarized in Table 2. For CAL, the highest values were found in BTA20 (0.21), followed by BTA6 (0.18). BTA21 (0.19), BTA4, and BTA9 (0.17) were the three autosomes that showed the highest extent of LD in MUP. For PON, BTA16 had the highest *r*^2^ (0.23), followed by BTA21 (0.22), BTA20, and BTA1 (0.20). BTA20 and BTA16 were also the chromosomes with higher LD extent for LIM, albeit at a much lower degree (0.13 and 0.12, respectively). The lowest LD was found in BTA22 (for PON and LIM) and in BTA27 (for CAL, MUP, and LIM). For LIM, the lowest LD values (0.05) were found in more BTA apart from BTA22 and BTA27 (Table 2).

The distribution of *r*^2^ across BTA is shown in Figure 2. The highest variation was observed for PON. On the contrary, LIM had both the lowest variation and the lowest *r*^2^ values. Median values were similar across BTA within breed except for PON, where more fluctuations were present. The reduced area of the first quartiles indicated that the observations had similar values and were close to 0. The third quartiles were wider than the first in all breeds. Moreover, the whiskers, representing maximum (and minimum) values, suggested the high differentiation of LD extent within the local breed and between the local and commercial breeds.

In order to analyze the LD decay within a specific distance between SNPs, independently of differences within chromosomes, LD was investigated considering intervals of 0.25 Mbp up to 1 Mbp (Figure 3). Differences among breeds were observed, especially for LIM, which was clearly separated from the local breeds with a steeper decay with an increasing distance.

PON and LIM presented the two opposite extremes (top and bottom on Figure 3, respectively) while CAL and MUP had similar and intermediate trends. For PON, an *r*^2^ close to 0.15 was maintained even at 1 Mbp distance, while for CAL and MUP, it was maintained at ~0.1. For LIM, *r*^2^
<0.1 was found for distances >0.12 Mbp (Figure 3). In general, for the local breeds, a sharp decay was observed till ~0.12 Mbp, while for LIM, the *r*^2^ was stabilized after ~0.25 Mbp. Average *r*^2^ for each bin was reported in Appendix A. Values of *r*^2^ > 0.2 in the first bin were found only for PON, while from 0.25 to 1 Mbp, values remained close to 0.15. MUP and CAL had similar trends, with *r*^2^ ~0.19 ± 0.25 in the first interval, whereas from 0.25 to 0.5 Mbp, CAL showed higher mean *r*^2^ value than MUP; while with the longest distance (>0.75 and up to 1 Mbp), *r*^2^ was close to 0.10 for both breeds. LIM showed the more rapid LD decay, with average *r*^2^ ranging from 0.141 (<0.25 Mbp) to 0.035 (>0.75 Mbp).

The *r*^2^ was also calculated per BTA in windows of 1 Kbp. The LD decay was found to be different among chromosomes, but with a similar trend in the local breeds (Appendix A). CAL and PON presented a higher variation of *r*^2^ within the analyzed interval of 1 Kbp; indeed, the decay did not linearly decrease, while LIM showed a faster decay.

### 3.3. Estimation of Historical and Contemporary Effective Population Size

On the basis of LD estimates, the trend of *N_e_* over time (per generation) for each breed was investigated (Figure 4). Similar to LD, the *N_e_* pattern of the local breeds was different from that of LIM. The decrease in *N_e_* was clear in all breeds, with a sharper decay for LIM. *N_e_* decreased from ~275 (80th generation ago) to 79 and 65 (13th generation ago) in CAL and MUP, respectively, while for PON, *N_e_* estimates decreased from 204 to 45 (in the most distant and recent generation, respectively). The historical trend of LIM was very different than those for the local breeds, decreasing from 920 to 310 (80th to 13th generations ago).

Regarding the contemporary *N_e_* (*cNe*), CAL, MUP, PON, and LIM showed values of 41.7, 18.7, 17.0, and 327.9, respectively.

## 4. Discussion

In this study, LD was investigated to assess the genomic architecture and the evolutionary history of autochthonous populations. LD provides various information that is used in several applications. For example, LD between linked markers determines the power and the precision of association mapping studies, because it influences the ability to localize genes, loci affecting economic traits, and diseases. Indeed, understanding the extent of LD improves the planning and the performance of genomic breeding programs [41]. However, direct comparison between the studies should be done with caution, owing to some factors that influence LD estimates, that is, sample size, population history and structure, LD measure (*r*^2^ or *D*′), marker type (microsatellites or SNPs), marker filtering, density, and distribution [34]. In the present study, a genome-wide LD extent and *N_e_* parameters were calculated for three Italian local beef breeds (CAL, MUP, and PON). The results were contrasted to the cosmopolitan Limousin, a commercial beef breed undergoing selection. This is the first study that performed an in-depth analysis of LD extent in these three Italian local breeds. In the last years, great importance has been given to genetic diversity and has underlined local breeds as important genetic resources as they harbor unique gene pools as a result of adaptation to the local environment [42]. In the absence of high-quality pedigree information, a GRM was constructed for each breed to define the degree of relatedness within a sampled breed. This information was used to investigate potential differences among the breeds that might be responsible for differences in LD and *N_e_* patterns. It is known that, in a closed inbred population, the recombination decreases and the LD increases [43]. The heatmaps showed that this situation has been avoided; farm groups were present, especially in CAL, but the relatedness within them was not found to be worrisome. Differences in LD and *N_e_* estimates of LIM compared with local breeds could be a result of different levels of relationship among the animals sampled, but, most of all, differences could be caused by the greater sample size and the breeding program used for LIM, which is absent for local breeds.

Regarding LD extent, our results revealed an *r*^2^ variation among BTA within the breed. This could be partly attributed to different lengths of the chromosomes [22]. However, BTA1, which was the longest chromosome, presented a high LD level only in PON. Interestingly, BTA16, BTA20, and BTA21 were characterized by high *r*^2^ values shared in more than one breed: BTA16 showed high *r*^2^ values in PON and LIM; BTA20 in CAL, PON, and LIM; and BTA21 in PON and MUP. These autosomes should be investigated in further works because the markers with high *r*^2^ values could reveal potential Quantitative Trait Loci (QTLs).

Average LD was different between local breeds and LIM. LIM is one of the most reared beef breeds in Italy, and was analyzed in this study to compare local breeds LD patterns with that belonging to a breed undergoing directional selection. In general, in commercial breeds, artificial selection and insemination allow the control of matings, and thus the inbreeding, which is further linked to LD, as inbreeding augments the covariance between alleles at different loci [9]. Furthermore, if the breed had an expansion in population size or consists of a large population, genetic drift is weaker and, as a consequence, LD decreases, converging to an equilibrium [43]. The higher LD observed in the local breeds is likely related to a higher ancestral relatedness and to a historically smaller *N_e_* [44], as shown in Figure 4. Moreover, the LD decay of the local breeds had the characteristic trend of populations that have suffered a collapse in population size and/or a bottleneck, as described by Rogers [43]. A previous study on Italian local cattle breeds carried out by Mastrangelo et al. [45] confirmed the slow LD decay found in this study. Mastrangelo et al. [44] analyzed two local breeds reared in Sicily, an island in the South of Italy. The *r*^2^ values were 0.16 for Cinisara and 0.20 for Modicana. No other studies on the level of LD in Italian local cattle were found in the literature. Hence, comparisons were done with foreign breeds. Mustafa et al. [46] investigated the Sahiwal dairy breed, which is under threat of extinction, showing an average *r*^2^ value equal to 0.18, which is similar to our Tuscan local breeds (ranged from 0.14 to 0.17). Nevertheless, differences were found for the LD decay, which was significantly more rapid than in Tuscan populations. Tunisian local breeds, consisting of big populations, studied by Jemaa et al. [22], showed more similar LD patterns to LIM than the Tuscan breeds, with *r*^2^ values lower than 0.05 in larger distances (>0.5 Mbp). Another study that corroborated the hypothesis that the decay of Tuscan breeds was slower than other local breeds was performed by Makina et al. [47], who studied four local African cattle compared with Angus and Holstein breeds. In all six breeds, at 1 Mbp, *r*^2^ values were lower than 0.1 and in the four local breeds (Afrikaner, Nguni, Drakensberger, and Bonsmara), and *r*^2^ did not exceed the value of 0.05 (in our study, all local populations had *r*^2^ > 0.1 at 1 Mbp distance).

More studies on LD on commercial beef cattle breeds can be found in the literature. For instance, in Chinese Simmental breed [48] in the window of 0.5–1 Mbp, the *r*^2^ was 0.05. This is similar to our estimates in LIM (0.03) and significantly lower than our findings in the local breeds (0.11 for CAL and MUP and 0.15 for PON). Biegelmeyer et al. [49] investigated the LD patterns in Hereford and Braford, and found average *r*^2^ values equal to 0.07 and 0.06, respectively, for the same window of 1 Mbp. Both breeds showed a great level of LD in the short term, suggesting a faster decay than in Tuscan breeds and more similar to LIM decay.

Other studies always based on selected breeds reported smaller average *r*^2^ by chromosomes for Limousin breed [15,50] than that found in the present research, likely owing to the different history of populations and to a more ancient breeding selection system compared with Italian Limousin.

Sample size is another factor that influences the estimate of LD. Khatkar et al. [51] declared that, if *r*^2^ parameter is used to calculate LD, a minimum sample size of 75 animals is required both for an accurate estimation and to avoid bias. LIM, as well as CAL and MUP, consisting of larger populations than PON, were sampled considering the threshold of the aforementioned study. In our analysis, only PON did not reach this threshold. However, this was because, at present, only 49 PON alive animals are officially enrolled to the register of cattle at limited diffusion. Hence, our samples referred to ~81% of the entire PON population. Regarding the two other local breeds, the genotypic data analyzed represented 47.5% and 65.4% of Calvana and Mucca Pisana, respectively. Overall, our study considered a much higher number of genotyped animals per breed compared with previous studies on autochthonous cattle breeds [7,16,20,22,23,45,52].

In fact, sample size could be the cause of the different results found in the present research when compared with previous studies. For example, Kukučková et al. [37] investigated the trend of historical *N_e_* of 15 European cattle breeds, Limousin included, and reported lower values (*N_e_* was equal to ~300 in the 60th generation ago) than the Italian LIM, but the sample size of that study was limited to 20 animals. Mastrangelo et al. [3] investigated the trends of *N_e_* of the same Tuscan breeds, among a plethora of cattle breeds, and related different trends of historical effective population size. The recent *N_e_* (13th generation) was less than 100, as in our study, but the trends for PON and MUP contrast with this latter with the lowest values and no overlapping with Calvana as instead related in the present study. Regarding LIM, both studies showed different trends between local and cosmopolitan breeds, but the LIM historical *N_e_* was higher in our study (80th generation *N_e_* was 920, whereas it was <500 in Mastrangelo et al. [3]). Obviously, the contemporary *N_e_* (*cNe*) also presented some differences, even maintaining the same ranking: *cNe* was greater in the present study for the Tuscan breeds, namely, 41.7, 18.7, and 17 (for CAL, MUP, and PON, respectively) and 33.5, 8.7, and 7.2, respectively, for the same breeds in the cited paper. Only LIM had lower *cNe* value (327.9 instead of 468.9), probably because of the sampling strategy (only animals of the last three generations were taken). However, CAL, and definitely MUP and PON, were at risk of loss of genetic diversity, presenting *N_e_* lower than 50 animals for generation, which is the threshold suggested by Food and Agriculture Organization (FAO) [53].

## 5. Conclusions

Our results, utilizing a medium-density SNP chip, demonstrated differences in LD and *Ne* patterns in four beef breeds, three locals under extinction (Calvana, Mucca Pisana, and Pontremolese) and one cosmopolitan (Limousin), reared in Italy. The greater genetic diversity loss, as dictated by the *N_e_* estimates, was found for Mucca Pisana. This work complements previous analysis carried out on pedigree information, both describing the population structure of the three local breeds. It is necessary to carefully monitor these populations and to strengthen the use of accurate mating plans in order to both increase the census size and control relatedness and inbreeding. LD results could be utilized in association studies as well as in the development of low-density SNP chip panels, for example, for parentage testing. Future studies could focus on the genomic regions with high *r*^2^ values, as well as investigate runs of homozygosity in the three local breeds.

## Figures and Tables

**Figure 1 animals-10-01034-f001:**
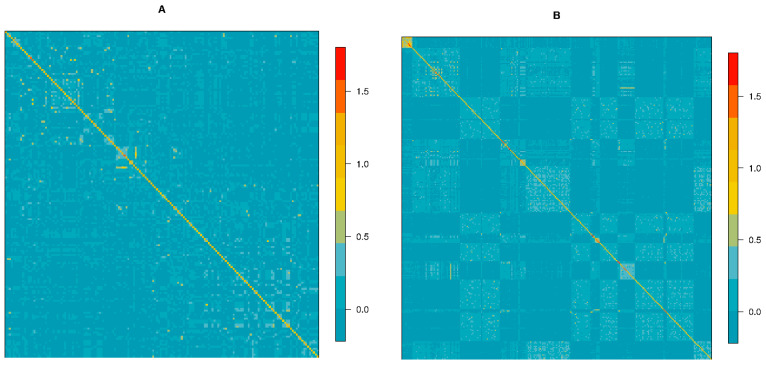
Heatmaps of genomic relationship matrix of (**A**) Calvana, (**B**) Mucca Pisana, (**C**) Pontremolese, and (**D**) Limousin breeds.

**Figure 2 animals-10-01034-f002:**
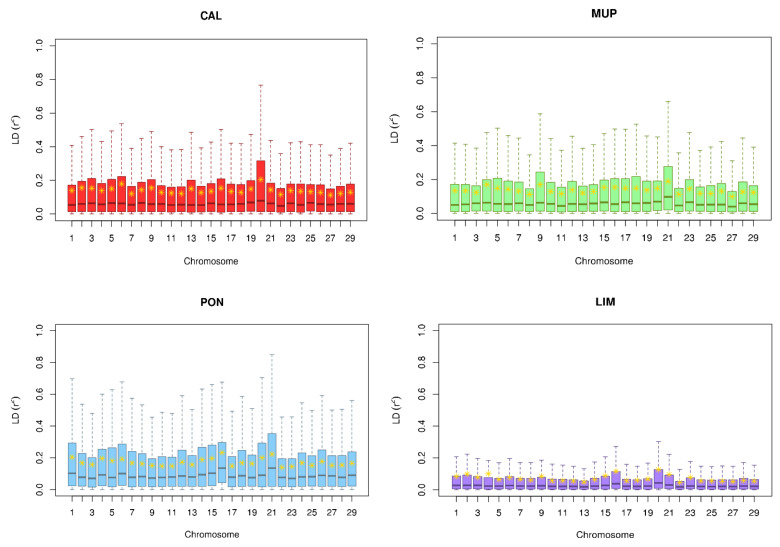
Boxplots of linkage disequilibrium (LD) (*r*^2^) of *Bos taurus* autosomes (BTAs) per breed. The yellow asterisks represent the mean and horizontal lines within each boxplot are the medians (outliers have been removed). CAL = Calvana; MUP = Mucca Pisana; PON = Pontremolese, LIM = Limousin.

**Figure 3 animals-10-01034-f003:**
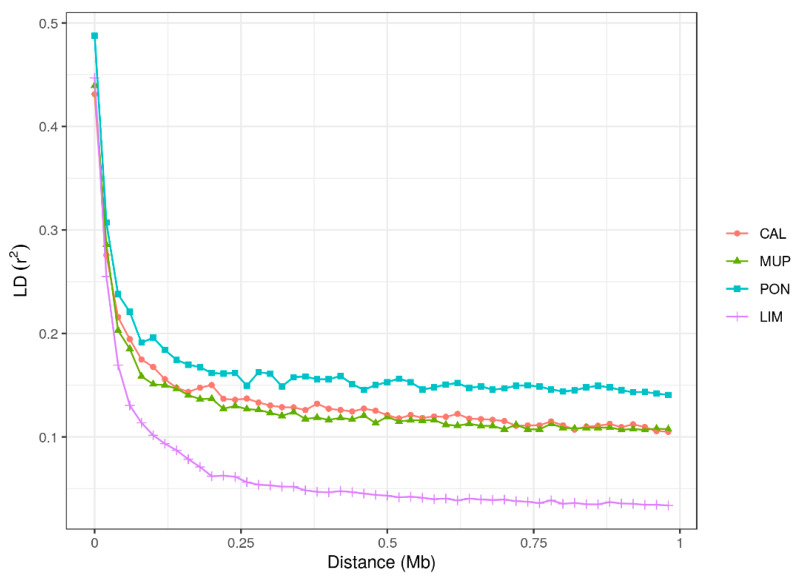
Linkage disequilibrium (LD), calculated as r^2^, in different distances of the genome and up to 1 Mbp for Calvana (CAL), Mucca Pisana (MUP), Pontremolese (PON), and Limousin (LIM).

**Figure 4 animals-10-01034-f004:**
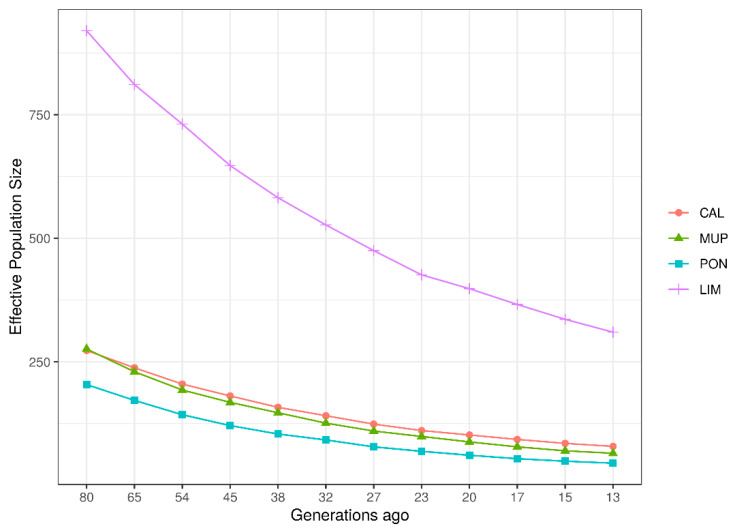
Average estimated historical effective population size (*N_e_*) in the Italian breeds: Calvana (CAL), Mucca Pisana (MUP), Pontremolese (PON), and Limousin (LIM).

**Table 1 animals-10-01034-t001:** Number of autosomal SNPs and individuals before (pre-) and after (post-)quality control (QC) per breed.

Breed ^1^	N SNPs Pre-QC	N SNPs Post-QC	N Individuals Pre-QC	N Individuals Post-QC
CAL	28,289	23,646	174	164
MUP	28,289	23,436	270	263
PON	28,289	22,791	44	41
LIM	28,289	23,279	100	100

^1^ CAL = Calvana; MUP = Mucca Pisana; PON = Pontremolese; LIM = Limousin.

**Table 2 animals-10-01034-t002:** The average and standard deviation (SD) of linkage disequilibrium (*r*^2^) for *Bos taurus* autosomes (BTAs) per breed.

Breed ^1^	CAL		MUP		PON		LIM	
Autosome	Average *r*^2^	SD	Average *r*^2^	SD	Average *r*^2^	SD	Average *r*^2^	SD
BTA1	0.14	0.20	0.13	0.20	0.20	0.25	0.08	0.16
BTA2	0.16	0.23	0.13	0.19	0.17	0.22	0.10	0.18
BTA3	0.15	0.21	0.12	0.17	0.16	0.22	0.08	0.15
BTA4	0.14	0.19	0.17	0.26	0.20	0.26	0.10	0.22
BTA5	0.15	0.20	0.15	0.20	0.18	0.23	0.07	0.12
BTA6	0.18	0.26	0.14	0.20	0.19	0.23	0.08	0.14
BTA7	0.12	0.16	0.13	0.17	0.17	0.21	0.07	0.12
BTA8	0.14	0.19	0.11	0.16	0.16	0.21	0.06	0.11
BTA9	0.15	0.22	0.17	0.23	0.15	0.21	0.08	0.17
BTA10	0.13	0.18	0.13	0.17	0.15	0.18	0.06	0.10
BTA11	0.12	0.18	0.12	0.16	0.15	0.18	0.06	0.11
BTA12	0.12	0.17	0.14	0.19	0.17	0.21	0.06	0.10
BTA13	0.15	0.21	0.12	0.17	0.16	0.20	0.05	0.09
BTA14	0.13	0.18	0.13	0.18	0.19	0.23	0.06	0.11
BTA15	0.14	0.18	0.15	0.21	0.20	0.24	0.09	0.17
BTA16	0.15	0.22	0.15	0.22	0.23	0.28	0.12	0.20
BTA17	0.13	0.18	0.15	0.20	0.15	0.18	0.06	0.10
BTA18	0.13	0.17	0.15	0.20	0.17	0.20	0.06	0.12
BTA19	0.15	0.20	0.14	0.19	0.16	0.21	0.07	0.13
BTA20	0.21	0.27	0.15	0.21	0.20	0.25	0.13	0.21
BTA21	0.14	0.21	0.19	0.24	0.22	0.25	0.10	0.18
BTA22	0.12	0.16	0.11	0.16	0.14	0.17	0.05	0.08
BTA23	0.14	0.20	0.14	0.19	0.15	0.19	0.08	0.15
BTA24	0.14	0.19	0.12	0.16	0.17	0.22	0.06	0.11
BTA25	0.13	0.17	0.12	0.16	0.15	0.19	0.06	0.10
BTA26	0.13	0.17	0.13	0.18	0.18	0.22	0.05	0.09
BTA27	0.11	0.15	0.10	0.14	0.15	0.18	0.05	0.09
BTA28	0.12	0.16	0.13	0.16	0.15	0.19	0.06	0.10
BTA29	0.15	0.22	0.12	0.17	0.20	0.25	0.05	0.09

^1^ CAL = Calvana; MUP = Mucca Pisana; PON = Pontremolese, LIM = Limousin.

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
