# Peer review of "Estimation of Linkage Disequilibrium and Effective Population Size in Three Italian Autochthonous Beef Breeds"

_animals, 2020, doi:10.3390/ani10061034_

Round 1

Reviewer 1 Report

The authors have revised the manuscript according to most of the comments. From what I remember, the major concerns were the PCA analysis and the Discussion section resembling a long literature review. I am not convinced with the answer to the first and the most important comment. The authors provide figures of 4 GRM, and mentioned that it is very analogous to conducting a PCA on a GRM. Well, it is not. They extracted a set of parameters from each GRM and together analysed in a PCA to represent a PCA on a multi-breed GRM. As I mentioned before, it is incorrect and misleading. The authors are encouraged to perform a PCA on a combined GRM of the 4 breeds. Please refer to this article as an example: “Genome-wide linkage disequilibrium and genetic diversity in five populations of Australian domestic sheep”.

Furthermore, heatmaps in Figure 1 should all follow the same colour scale (~-0.2 to ~1.6).

Author Response

AU: We thank the reviewers for their review. Please find below our responses to the points raised. All our responses are preceded by “AU”. Changes in the manuscript are done in track changes and highlighted in yellow. We hope to find the new version of the manuscript suitable for publication in the “Animals” journal.

Looking forward to hearing from you.

Sincerely,

Maria-Chiara Fabbri

The authors have revised the manuscript according to most of the comments. From what I remember, the major concerns were the PCA analysis and the Discussion section resembling a long literature review. I am not convinced with the answer to the first and the most important comment. The authors provide figures of 4 GRM, and mentioned that it is very analogous to conducting a PCA on a GRM. Well, it is not. They extracted a set of parameters from each GRM and together analysed in a PCA to represent a PCA on a multi-breed GRM. As I mentioned before, it is incorrect and misleading. The authors are encouraged to perform a PCA on a combined GRM of the 4 breeds. Please refer to this article as an example: “Genome-wide linkage disequilibrium and genetic diversity in five populations of Australian domestic sheep”.

Furthermore, heatmaps in Figure 1 should all follow the same colour scale (~-0.2 to ~1.6).

AU: We are convinced that PCA can be conducted on any set of variables and also in this case, PCA analysis would be useful to give a general picture of these breeds. However, The PCA has been removed from the manuscript.

The heatmap colors has been modified as requested.

Reviewer 2 Report

I think the authors made a strong effort to consider my comments.

They have obviously not understood my remark about PLINK.

They use two different spellings. This should be harmonized.

The official version seems to be "PLINK".

Author Response

AU: We thank the reviewers for their review. Please find below our responses to the points raised. All our responses are preceded by “AU”. Changes in the manuscript are done in track changes and highlighted in yellow. We hope to find the new version of the manuscript suitable for publication in the “Animals” journal.

Looking forward to hearing from you.

Sincerely,

Maria-Chiara Fabbri

I think the authors made a strong effort to consider my comments.

They have obviously not understood my remark about PLINK.

They use two different spellings. This should be harmonized.

The official version seems to be "PLINK".

AU: Authors are apologized to not understand what the review was underlining. In the abstract, “plink” has been changed to “PLINK”, as it is called in Materials and Methods.

Reviewer 3 Report

The authors addressed all issues proposed by all reviewers and thus I recommend to accept the revised manuscript for publication.

Author Response

The authors addressed all issues proposed by all reviewers and thus I recommend to accept the revised manuscript for publication.

AU: Authors thank for the review.

Round 2

Reviewer 1 Report

The authors have done a major revision, and the manuscript has improved considerably. There are minor issues with the manuscript that requires correction.

L42: "PON faces nowadays" to "PON faces"
L58: "usually is" to "it is"
L79: "the gene mapping" to "gene mapping"
L80: "three beef cattle" to "three Italian beef cattle"
L91: "and balanced by sex" to "with equal number of male and female samples" ?
L135: "Default options" to "The default options"
L136: "MAF was set to > 0.01". You did not set MAF to > 0.01. You considered alleles with MAF > 0.01. This sentence is repeated from L98. Pleasde delete this sentence.
L137: "13 generations vs. 4000 Kbp" needs clarification.
L144: "by [39]" to "by Ohta and Kimura [39]"
L145: "the NEESTIMATOR" to "NEESTIMATOR"
L146: "and the lowest allele frequency used to 0.01". It is repeated a few times before.
L185-186: "0.104, 0.105, 0.106 ± 0.08 Mbp in CAL, MUP and PON, LIM respectively". 3 values for 4 breeds! Please be more cautious with mistakes after a few rounds of revision.
L205: Delete "and was intermediate for CAL and MUP".
L207: "fluctuations were present" to "more fluctuations were present"
L207-209: Delete these lines. You don't need to repeat the whole figure or the table completely in the text.
L222: "More precisely" compared to what? Please delete it.
L222: "PON and LIM presented the two opposite extremes" Are you sure that both of them extremes?! You may say, one showing the highest, and the other showing the lowest.
L241: "Similarly" to "Similar"
L243: "while for PON Ne" to "while for PON, Ne"
L245: "historical trend of LIM resulted extremely different than the local breeds" to "historical trend of LIM was very different than those for the local breeds"
Discussion: In a previous comment, I asked the authors to shorten the Discussion. This time, I skipped reading it. Whether 53 references were needed for this study, I don't know. The authors should consider that they are writing for the readers to read not to skip.

Author Response

AU: We thank the reviewer for the constructive review. Please find below our responses to the points raised. All our responses are preceded by “AU”. Changes in the manuscript are done in track changes. We hope to find the new version of the manuscript suitable for publication in the “Animals” journal.

Looking forward to hearing from you.

Sincerely,

Maria-Chiara Fabbri

The authors have done a major revision, and the manuscript has improved considerably. There are minor issues with the manuscript that requires correction.

L42: "PON faces nowadays" to "PON faces"

AU: Changed as suggested (L43)

L58: "usually is" to "it is"

AU: Changed as suggested (L60)

L79: "the gene mapping" to "gene mapping"

AU: Changed as suggested (L82)

L80: "three beef cattle" to "three Italian beef cattle"

AU: Changed as suggested (L83)

L91: "and balanced by sex" to "with equal number of male and female samples" ?

AU: the number of males and females has been added to the manuscript (L94)

L135: "Default options" to "The default options"

AU: Changed as suggested (L144)

L136: "MAF was set to > 0.01". You did not set MAF to > 0.01. You considered alleles with MAF > 0.01. This sentence is repeated from L98. Pleasde delete this sentence.

AU: Changed as suggested (L144-145)

L137: "13 generations vs. 4000 Kbp" needs clarification.

AU: These parameters are reported by the software and the cited authors

L144: "by [39]" to "by Ohta and Kimura [39]"

AU: Changed as suggested (L153)

L145: "the NEESTIMATOR" to "NEESTIMATOR"

AU: Changed as suggested (L155)

L146: "and the lowest allele frequency used to 0.01". It is repeated a few times before.

AU: Changed as suggested

L185-186: "0.104, 0.105, 0.106 ± 0.08 Mbp in CAL, MUP and PON, LIM respectively". 3 values for 4 breeds! Please be more cautious with mistakes after a few rounds of revision.

AU: Values have been corrected. However, CAL and MUP presented the same value (0.105), see Table S2

L205: Delete "and was intermediate for CAL and MUP".

AU: Changed as suggested (L232)

L207: "fluctuations were present" to "more fluctuations were present"

AU: Changed as suggested (L234)

L207-209: Delete these lines. You don't need to repeat the whole figure or the table completely in the text.

AU: The sentence has been reduced (L234)

L222: "More precisely" compared to what? Please delete it.

AU: Changed as suggested (L252)

L222: "PON and LIM presented the two opposite extremes" Are you sure that both of them extremes?! You may say, one showing the highest, and the other showing the lowest.

AU:  The meaning of the sentence is that the two breeds are at the opposite extremes in the graph. Authors prefer to leave the sentence as it is in order to not repeat “high” and “low” expressions.

L241: "Similarly" to "Similar"

AU: Changed as suggested (L274)

L243: "while for PON Ne" to "while for PON, Ne"

AU: Changed as suggested (L278)

L245: "historical trend of LIM resulted extremely different than the local breeds" to "historical trend of LIM was very different than those for the local breeds"

AU: Changed as suggested (L280)

Discussion: In a previous comment, I asked the authors to shorten the Discussion. This time, I skipped reading it. Whether 53 references were needed for this study, I don't know. The authors should consider that they are writing for the readers to read not to skip.

AU: Authors thank the reviewer. We improved the discussion, but we prefer to leave the included References to allow to examine in depth the specific topics for interested readers

Reviewer 2 Report

the authors have addressed duly my point of criticism

such that I recommend the publication of the manuscript.

Author Response

AU: We thank the reviewer for the constructive review.

This manuscript is a resubmission of an earlier submission. The following is a list of the peer review reports and author responses from that submission.

Round 1

Reviewer 1 Report

The manuscript is about the study of LD and effective population size in 3 local Italian cattle breeds and comparisons with Limousin breed. This is an interesting study. However, there are key points that need to be addressed.

1. The PCA analysis is incorrect. A PCA analysis based on descriptive statistics of the GRM does not have any information. What you need to do is to perform PCA on the GRM itself.
2. The English grammar and the scientific writing should defenitely be improved. The manuscript is full of mistakes, and unfortunately, it is not written carefully.
3. There are long sentences in the manuscript, some sentences are not necessary. Keep it short and get to the point.
4. In the results section, authors tend to repeat the tables and figures in the text.
5. The discussion section looks like a literature review report rather than a discussion section. Some of the literature review can be moved to Introduction. Please discuss your findings and report relevant findings from the literature. The reader is interested to know about your findings, not reading a lengthy literature review.

Please see below for further comments.

Use capital letters in the title (e.g., beef -> Beef)
L18: "control" to "control group"
L20: "genotypes" to "genotype"
Cange missingness to call rate.
"r2" and "Ne" should be italic everywhere.
"contemporary" to "current" everywhere
L24: "was found" to "was"
L24: "Calvana e" to "Calvana and"
L28: "across generations" to "across generations for local breeds"
L30,60: "demographical" are you sure that this is the right word?!
L37: "basin" ???
L39: "(Calvana, CAL; Mucca Pisana, MUP; and Pontremolese, PON)"to "(Calvana (CAL), Mucca Pisana (MUP) and Pontremolese (PON))"
L42: "Pontremolese faces nowadays the" to "PON faces nowadays a"
L43: "of only few" to "a limited number of"
L43: "the breed" which breed?
L45: "comes" to "originates"
L51: "the region" to "this region"
L54: "the size of an ideal population" to "a population size". There is nop ideal population or ideal population size!
L55: "change as the real population under study" to "changes as a real population"
L60: "the general" to "generally"
L66: "of alleles" to "between alleles"
L66: The sentence needs a reference.
L88: "analyzed" to "genotyped"
L92: "Limousin" to "Limousin, City". Mention the city where ANACLI is located.
L94: Use capital letter only for the first letter. The same for other headings.
L99 and elsewhere: "<" to " < " and ">" to " > "
L104: "relationships" to "identical by state relationships"
L104: "status" which status?
Eq1,L107,L108: bold Z
L107: italic "pi"
L108: "Matrix Z" to "Z"
L109: "matrix X which" to "the matrix that"
L111: "For each GRM" to "For the GRM of each breed"
L112 and elsewhere: "off diagonal" to "off-diagonal" and "diagonal" to "diagonal values"
L112: "values of the diagonal" to "of diagonal values"
L123: "values of the off diagonal" to "off-diagonal values"
L124: "absolute and squared root" of what?
L116: Please refer to my previous comment on the PCA.
L120: "better measure of LD because" to "better measure of LD than D', because"
L121: "change" to "changes"
L121: delete "as D'"
L121: "The r2 values range" to "The r2"
L122: delete this line
L123: Change it to "was calculated as:"
Eq2: Change freq X to freq(X) and * to x.
L125-126: Change them to "where freq(A), freq(a), freq(B) and freq(b) are the allele frequencies and freq(AB), freq(ab), freq(Ab) and freq(aB) are the genotype frequencies. The LD extent was"
L127: "in PLINK" to "using PLINK"
L128: More description is needed here.
L129: "Also, the" to "The"
L130: "Mbp: 0 - 0.25; 0.25 - 0.5; 0.5 - 0.75; 0.75 - 1 Mbp" to "Mbp (0 - 0.25, 0.25 - 0.5 0.5 - 0.75 and 0.75 - 1 Mbp)"
L131: "had been" to "were" and "has been" to "was also"
L132: delete "also"
L132 and elsewhere "kb" to "Kbp"
L136: "recent effective population size" to "current Ne". After using an abbreviation, do not use the full term!
L137: "relationship" to "relationships"
L140: "4000kb" to "4000 Kbp"
L141: "50kb" to "50 Kbp"
L142: "4000 kb" to "4000 Kbp"
Eq3 and elsewhere: "NT(t)" to "Ne(t)"
L144: "estimated t generation ago" to "estimated for t generation ago". "t" should be italic.
L145: "Ct" to "ct" and "rate t" to "rate at t"
L147 and elsewhere: "cNe" to "Ne0"
L148: "v. 2" to "v.2" and "set on random option" to "set to random"
L154: "individuals for each breed" to "individuals from each breed"
L157-158: Change to "Table 1. Number of autosomal SNPs and individuals before (pre-) and after (post-) quality control (QC) per breed."
Table 1: Delete "Local" and "Commercial" rows.
L162: "The lower" to "The"
L164: "Calvana." to "CAL"
L165: "relatedness" to "relatedness average"
l173-176: Change to "<1 in all breeds. The average of the diagonal values was 0.99 for CAL, MUP and LIM, and 0.97 for PON. The highest diagonal values were 3.25, 1.75, 1.54 and 1.22 for PON, MUP , CAL and LIM. The minimum diagonal values ranged from 0.67 (MUP) to.0.78 (LIM)." I think the sentences are unnecessarily too long.
L177: "in all breeds" to "for all breeds"
L178-188: Delete these lines and replace them with the new PCA results.
L194-195: When you mention a range, mention it from small to large not the other way around.
L198: "namely" ?
L200: "The highest average" to "The average of"
L200-201: Why ~0.17 and ~0.14? Either 2 or 3 decimals.
L201: "The mean" to "The mean and SD of"
L202: "Table 2" not "Table 1". The authors have not written the manuscript carefully.
L202: "= 0.21" to "0.21"
L203: "r2 = 0.19" to "0.19" and "r2 = 0.17" to "0.17"
L202-208: Re-write this paragraph. Difficult to read.
L210: Change it to "Table 2. The average and standard deviation (SD) of linkage disequilibrium (r2) for Bos Taurus"
Table 2: "CAL1" to "CAL". Add headers "Breed1" and "Autosome"
L218: "values" which values?
L218: "the majority of which was <0.2" to "within breed"
L218: "Medians" to "Median"
L219-223: Re-write thse lines. Long and unclear. Write it short, clear and concise please.
L228: "decay behavior" to "decay"
L230-232: Change it to "Mbp and up to 1 Mbp (Figure 4). Different breeds showed different patterns of LD decay."
Figure 4: "LD (r2)" to "Linkage disequilibrium (r2)"
L234: "LD decay plot" to "Linkage disequilibrium in different distances of the genome"
Delete the first sentence in lines 236-237. It is unnecessary.
L238: "maintained" to "remained". Please check the English throughly in the manuscript.
L238: "it was at" to "it was remained at"
L239: ">0.12 Mbp" to "> 0.12 Mbp (Figure 4)"
L240: "was" to "is"
L241-246: Long and difficult to read. Please re-write it.
L243: "higher value". You need to mention which value!!! Mean or SD?
Table 3: "Cal1" to "CAL". Add a header "Breed1"
L250 "1 kb" to "1 Kb"
L252: "1 Kbp" to "1kb"
L253: "more uniform and rapid" to "faster"
L256: "breed, was" to "breed was"
L258-260: Delete "None of the local breeds had an estimated Ne higher than 100 in the 13th generation. In general, CAL had a slightly higher Ne through generations, compared to MUP and PON. More precisely," Very wordy and repeating what is shown in the figure.
L262: "varied between 204 and 45 (in the most distant and recent generation, respectively)." to "decreased from 204 to 45 (80th to 13th generations ago)"
L263-264: "with a maximum of 920 (80 generations ago) and a minimum of 310 on generation 13th." to "decreasing from 920 to 310 (80th to 13th generations ago)"
Figure 5: "Effective Population Size" to "Effective population size"
L268-270: Change to "Regarding the current Ne (Ne0), CAL, MUP, PON and LIM showed 41.7, 18.7, 17.0 and 327.9, respectively."
L272: "effective population size parameters" to "Ne"
L274: "a beef breed" to "a commercial beef breed"
L279: delete "more precisely, "
L280-286: Update these lines with the new PCA results.
L292: Change to "Average LD was different between local breeds and LIM. LIM is one"
L299: "in an equilibrium" to "to an equilibrium"
L299: "observed in PON, but also in the other two local breeds" to "observed in the local breeds"
L301: "belonging to" to "of"
L302: "had the characteristic curve of populations" ???
L304: "confirmed the slower LD decay found in this study" It is not clear what you mean. Slower than what? Re-write it.
L304: "The authors analyzed two local breeds" to "Mastrangelo et al. [44] analyzed two local breeds Cinisara and Modicana"
L305: "an island of Italy, and for these breeds the" to "an island in Italy. The"
L23306: "Modicana, similar to PON (0.17) but higher than CAL and MUP (0.14)." to "Modicana." I deleted the rest, because the values are not similar.
Because of the huge amount of edits, I stop typing here. Please revise the manuscript throughly.
There are also mistakes in the references. See L455 and L460 for example.

Reviewer 2 Report

Estimation of LD and Effective Population Size in three Italian autochthonous beef breeds

by Fabbri et al.

Many local beef breeds are endangered; carrying out research into them is commendable and deserves support.

The authors were investigating the genomic architecture of three local Italian breeds and were especially interested in the effective population sizes and how they change with time.

The analysis seems relatively sound, but the presentation of the results is not very appealing.

Major points are:

    •  

How exactly were the r squared values adjusted (formula 3)? Which value was chosen for alpha? And why? Alpha is rather a parameter than a constant and should be explained briefly.

    •  

Figure 1: the colors are difficult to compare since the scales are different (maximum values from ~ 1.2 (D) to ~1.6 (A). Should be consistent.

    •  

Table 2: this is tiring, should be omitted or go to the Supplemental Material

    •  

Figure 2: an ordination plot (e.g. MDS) showing each and every animal, with colors according to breed, would be more conclusive.

    •  

Table 3: given that the changes of r squared are happening almost exclusively in the first interval, this table and the corresponding analysis does not make sense to me.

    •  

Figure 3: this is again quite meaningless. I don‘t see the point to show each and every chromosome when they are more or less the same within each breed. Think about a different presentation.

    •  

Supplemental tables: what is the interest of S1? Why is the chromosome length differ between breeds in S2? S3 needs a caption. S2 needs a caption with more details explaining the variables.

    •  

Discussion: Line 280: „A PCA … not only differentiated the local breeds from the commercial...“ This is not justified at all. You could state this if you had found that the three local breeds form a group with LIM being apart. This is obviously not the case.

    •  

Explaining differences between and similarities of effective population curves with relatedness between the populations does not make sense. You could have two identical curves from two species that are completely distinct and vice versa.

The text is full of carelessness and imprecision.

Examples are:

    •  

Ne instead of N index e

    •  

Line 28: „sample size“ instead of effective population size

    •  

Spelling of PLINK 1.90

    •  

Line 145: Capital C instead of small c

    •  

Line 236/7: „Figure 3“ instead of Figure 4

    •  

Table 3: horizontal line below column headers too short

    •  

Line 454/5: The journal is called „J Anim Breed Genet.“

Reviewer 3 Report

The manuscript by Fabbri and colleagues reported a study on linkage disequilibrium (LD) among three Italian local cattle breeds (Calvana, Mucca Pisana ,Pontremolese) using Limousin breed as control. The results showed that Calvana and Mucca Pisana breeds had moderate level of LD (~ 0.14) and Pontremolese had the highest level of LD (0.17), whereas Limousin presented the lowest level of LD (0.07). The obtained results would provide scientific evidence for conservation of three local cattle breeds with very low effective population sizes (Ne). In general, the manuscript is well-organized and well-written. Minor changes need to done before final acceptance.

  1. In lines 202-204, BTA20 (=0.21), BTA6 (=0.18), BTA1 (=0.20) should be better changed to BTA20 (0.21), BTA6 (0.18), BTA1 (0.20) .
  2. The Ne should be italicized in the manuscript, such as the first and second paragraphs on page 10.
  3. The sentences and punctuation need to be checked and corrected again in the manuscript. For example, in lines 342-343, related different trends of historical effective population size; in line 24, "Calvana e Mucca Pisana" should be "Calvana and Mucca Pisana".
  4. Why the sample size of Pontremolese (n = 44) was much lower than those of other breeds. Would the sample size bias the estimations of LD and Ne for this breed in comparion with other breeds?
